# An Optimization Method of Urban Rail Train Operation Scheme Based on the Control of Load Factor

**Fei Dou** [1,2], **Huiru Zhang** [1,2], **Haodong Yin** [3,*], **Yun Wei** [1,2] **and Yao Ning** [1,2]

1   Beijing Mass Transit Railway Operation Corp. Ltd., Beijing 100044, China
2   Beijing Key Laboratory of Subway Operation Safety Technology, Beijing 100044, China
3   State Key Laboratory of Rail Traffic Control and Safety, Beijing Jiaotong University, Beijing 100044, China
*   Correspondence: hdyin@bjtu.edu.cn

**Abstract:** The train operation scheme of urban rail transit is a transportation plan formulated to fully meet the needs of passenger travel under the constraints of signal system capacity, turn-back capacity, and so on. Facing an unexpected epidemic, it was particularly important for passengers to travel safely and in an orderly manner. With an ever-increasing passenger flow due to work resumption, this paper proposes an optimization method for the urban rail train operation scheme based on the control of the target load factor according to the preparation process of the train operation scheme. The proposed method obtained the optimal train running interval and routing scheme based on analyzing the spatiotemporal distribution of passenger flow. The north section of Beijing Subway Line 8 was taken as an example. After optimization, for trains in the morning peak hour in the downward direction, the maximum load factor for the collinear section of the full-length routing and short-turn routing was reduced by 21%, and the matching effect of the transportation capacity and volume in the non-collinear was improved. In general, the maximum load factor in the downward direction after optimization was 80%, which met the target control requirements. The results show that the optimization method plays an important role in balancing the load factor in each cross-section and realizing the optimal coupling of passenger flow and train flow.

**Keywords:** urban rail transit; train operation scheme; optimization method; load factor

## 1. Introduction

Facing an unexpected epidemic, it was particularly important for passengers to travel safely and in an orderly manner. With the ever-increasing passenger flow due to work resumption, controlling the load factor will effectively ensure the safety of passengers under the condition of the city's strict epidemic prevention requirements. The optimization of traffic organization has become an indispensable research method to relieve the pressure of passenger flow and further improve the transportation capacity of urban rail transit under the constraints of line signal system capacity [1], power supply system capacity [2], turn-back capacity [3] and so on.

Given the passenger flow characteristics of urban rail transit lines, scholars have carried out a wealth of theoretical research from the perspective of traffic organization. Considering that increasing the operation speed will help to reduce the running and waiting time, thereby improving the service level, Freyss et al. [4] design a skip-stop strategy that effectively reduced the travel time by reducing the time required for stopping at stations. To meet the demand of passenger flow with uneven spatial and temporal distribution, Tang and Xu [5] proposed a train plan combining full-length and short-turn routing with express-local modes. Similarly, to meet the needs of passengers, Dai et al. [6] adopted the full-length and short-turn routing scheme, Huang et al. [7] considered the express-local mode, and Rong et al. [8] studied hybrid train formation. For urban rail transit lines with multiple depots, Zhang et al. [9] considered the short turning strategy, and established a mixed integer nonlinear programming model, which is verified based on the

data of Beijing Subway line 4. For the intercity rail lines, Niu and Zhang [10] established an optimization model for the operation scheme of multi-group trains, to minimize the waiting time of passengers and the in-train crowded costs. Considering the time-varying passenger demand and predetermined train skip-stop patterns, Niu et al. [11] proposed a unified quadratic integer programming model with linear constraints to synchronize effective passenger loading time windows and train arrival and departure times at each station, so that the total waiting time of passengers at stations is minimized.

In terms of train operation optimization, some scholars have evaluated and optimized the train operation scheme based on the production benefits of operating companies and passenger travel services. Canca et al. [12] adopted a short-turning strategy to achieve the objective of diminishing passenger waiting time while maintaining a certain level of service. Based on the calculation of the operational cost of urban rail transit company and the travel cost of passengers belonging to different categories, Yao Enjian et al. [13] formulated a train operation optimization model using a long-short route strategy, which aimed at maximizing transport capacity and minimizing the passenger's travel delays. Xu Dejie et al. [14] proposed a generalized passenger fare calculation method under two kinds of train operation proportion in an urban rail transit system, and established an optimization operation scheme model of hybrid formation of full-length and short-turn routings to minimize passenger travel expenses and enterprise operating express. Based on constraints such as train marshalling, minimum time interval, and occupancy rate, Ma Caiwen et al. [15] established a nonlinear dual-objective integer programming model to minimize passenger travel costs and business operating costs. Taking into account the impact of cross-line trains on the carrying capacity of rail lines, Yang Anan et al. [16] formulated a capacity allocation model, which aims to minimize the costs, including rolling stock usage, train operating and passenger travel, under the constraints of departure interval, full-load rate, and the number of available rolling stock. Considering the spatial disparity in the utilization rate of the line conveying capacity, Shi Ruijia et al. [17] used the routing plan, service frequency, and formation plan as the decision variables, and designed an optimization model of the train plan to maximize the line conveying capacity utilization, minimizing the used train number and passenger travel cost. In addition, minimizing train energy consumption [18,19], train running times [18], or passenger waiting times [19,20] have also been studied by some scholars.

The above studies are either based on a given routing scheme [13,15], or compare the performance indicators of several determined routing schemes to obtain the best one [17], which indicates that the research on the routing scheme is not deep enough. In addition, in the existing research, most scholars aim at balancing the load factor in different periods to achieve a match between passenger flow and transport capacity [21,22], while few studies focus on the control of the load factor. These theoretical studies have laid a solid foundation for the optimization of train operation plans. However, limited by strong model assumptions and unspecific constraints, it is difficult to fully apply the theoretical research results to on-site train operations, especially for scenarios with target load factor constraints during major emergencies. Therefore, by analyzing the spatiotemporal distribution of passenger flow, this paper proposed an optimization method for the urban rail train operation scheme based on the control of load factor to match the transportation capacity and volume. This paper aimed to provide a method support for normalized transportation organization, and also to provide a reference for improving the level of operation service and the construction and operation of smart urban rail.

The rest of this paper is organized as follows. In Section 2, we introduce the compilation strategy of the train operation scheme and define the time imbalance coefficient and spatial imbalance coefficient to describe the distribution of passenger flow. An optimization model of the train operation scheme is described in Section 3. Based on the alternative routing set, Section 4 describes the algorithm for solving the routing scheme of train operation. A case study based on real-world data from the north section of Beijing Subway Line

8 is described in Section 5 to verify the effectiveness of the model and algorithm. Finally, conclusions are given in Section 6.

## 2. Strategies for Compiling Train Operation Scheme

Historic passenger flow data or predicted passenger flow data [23,24] analyzed the temporal and spatial distribution characteristics of passenger flow, such as section passenger flow, entrance/exit passenger flow, and transfer passenger flow to obtain the disparity of passenger flow in time and space. On this basis, the train operation interval, routing scheme, and skip-stop patterns in different time periods are calculated to realize the precise matching of transportation capacity and volume in the whole-day operation organization. According to the basic conditions of line equipment and facilities, and the impact of transfers between lines on train operation, an optimized train operation plan is finally formed. The flow chart for the preparation of the train operation scheme is shown in Figure 1.

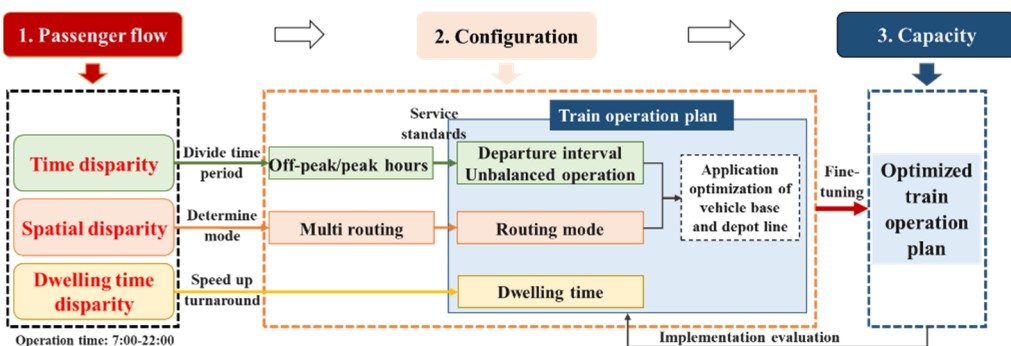

**Figure 1.** The process for compiling train operation scheme.

### 2.1. Analysis of Time Imbalance of Passenger Flow

The time imbalance coefficient is used to evaluate the distribution of passenger flow in time. The time imbalance coefficient refers to the ratio of the hourly maximum section passenger flow volume in one direction to the average value of the time-sharing maximum section passenger flow volume in that direction. The formula is as follows:

$$\mu_t^s = \frac{q_{t,\max}^s}{\sum\limits_t q_{t,\max}^s / T} \tag{1}$$

where $\mu_t^s$ is the time imbalance coefficient in time period $t$ in direction $s$, $q_{t,\max}^s$ is the maximum section passenger flow volume in time period $t$ in direction $s$, and $T$ is the operation time of the line.

According to the book *Urban Rail Transit Planning and Design* [25], if $\mu_t^s < 1.5$, the passenger flow is considered to be relatively balanced; otherwise, the passenger flow is considered to be imbalanced. Therefore, the peak and off-peak hours are determined based on the time-sharing section passenger flow volume. The analysis process of the time imbalance of passenger flow is shown in Figure 2.

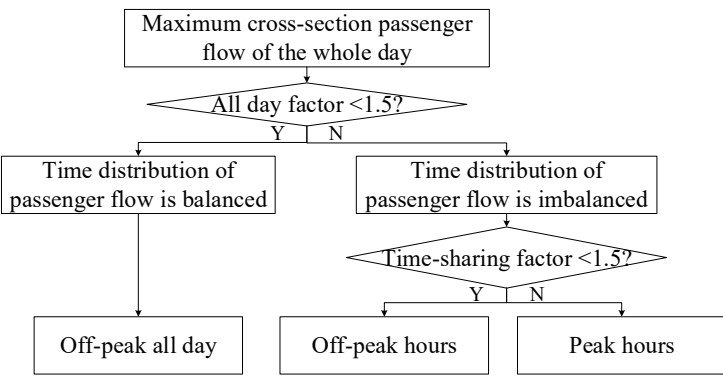

**Figure 2.** Analysis process of time imbalance of passenger flow.

*2.2. Analysis of Spatial Imbalance of Passenger Flow*

The spatial imbalance coefficient is used to evaluate the distribution of passenger flow in space. The spatial imbalance coefficient refers to the ratio of the maximum section passenger flow volume in one direction to the average value of all the maximum section passenger flow volume in that direction within a certain period of time. The formula is as follows:

$$\mu_e^s = \frac{q_{e,\max}^s}{\sum\limits_e q_{e,\max}^s / E} \tag{2}$$

where $\mu_e^s$ is the spatial imbalance coefficient of the cross-section $e$ in direction $s$, $q_{e,\max}^s$ is the maximum section passenger flow volume of the cross-section $e$ in direction $s$, and $E$ is the total number of sections.

According to the book *Urban Rail Transit Planning and Design* [25], if $\mu_e^s$ tends to 1, the section passenger flow volume is considered to be relatively balanced, and if $\mu_e^s \geq 1.5$, the section passenger flow volume is considered to be imbalanced. Therefore, the routing scheme is determined by $\mu_e^s$. If $\mu_e^s \geq 1.5$, the full-length and short-turn routings can be adopted, otherwise, a single full-length routing is recommended. Then, on the basis of the obtained routing scheme, the turn-back station of the short-turn routing can be determined according to the line conditions. The analysis process of spatial imbalance of passenger flow is shown in Figure 3.

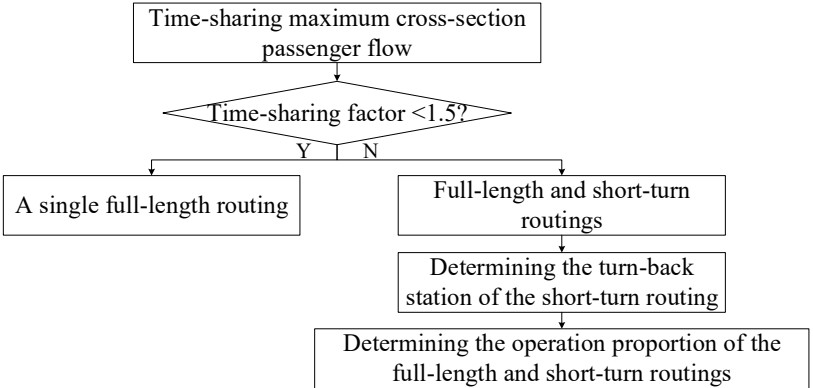

**Figure 3.** Analysis process of spatial imbalance of passenger flow.

When determining the turn-back station of the short-turn routing, in addition to the line conditions, relevant factors such as the load factor and headway should also be comprehensively considered. The turn-back station of the short-turn routing should be set as far as possible at the station where the cross-section unbalance coefficient in both directions is less than 1.5. Moreover, the principles of meeting the target load factor and reducing the complexity of passenger organization should be followed.

### 3. Optimization Model of Train Operation Scheme

According to the strategies for compiling the train operation scheme, an optimization model of the train operation scheme is established to match the transportation capacity and volume under the given load factor.

*3.1. Objective Function*

The goal of this paper is to match the transportation capacity and volume as much as possible, that is, the smaller the deviation between capacity and volume, the better. Therefore, the objective function is:

$$\min Z = \left| C \cdot L_t \cdot N_t - q_{t,\max}^s \right| \tag{3}$$

*3.2. Constraints*

1. Train running interval constraint;

$$\begin{cases} h_{\min} \leq h_t^s \leq \max\left\{ T / \frac{q_{t,\max}^s}{C \cdot L_t}, h_{\min} \right\}, peak\ hours \\ \min\left\{ T / \frac{q_{t,\max}^s}{C \cdot L_t}, h_{\max} \right\} \leq h_t^s \leq h_{\max}, off - peak\ hours \end{cases} \tag{4}$$

In general, the maximum headway of trains during off-peak hours is 600 s. Here, the minimum headway of trains mainly considers the capacity constraints of the power supply system and the signal system, namely:

$$h_{\min} \geq \max\left( h'_{\min}, h''_{\min} \right) \tag{5}$$

2. Maximum cross-section load factor constraint;

The maximum cross-section load factor in time period $t$ must meet the following conditions:

$$L_t = \frac{q_{t,\max}^s}{C \cdot (T / h_t^s)} \leq L_{\max} \tag{6}$$

3. Total number of trains;

$$\sum_r k_r \leq K \tag{7}$$

where $k_r \in Z$. This formula constrains the number of trains used in the optimized scheme to be no greater than the total number of trains in the line.

4. Distance constraint for short-turn routings;

The running distance of the short-turn routing should be suitable. If the distance is too short, the train will frequently turn back and clear passengers, thereby affecting the service quality. If the distance is too long, it is difficult to achieve the purpose of reducing cost and increasing efficiency by adopting short-turn routing. Therefore, the distance must satisfy the following constraints:

$$Z_{\min} \leq S_{\text{end}} - S_{\text{start}} + 1 \leq Z_{\max}, 1 \leq S_{\text{start}} < S_{\text{end}} \leq N \tag{8}$$

When $N \geq 25$, $Z_{\min} = (1/4) \cdot N$, $Z_{\max} = (3/4) \cdot N$; otherwise, $Z_{\min} = (1/3) \cdot N$, $Z_{\max} = (3/4) \cdot N$.

5. Load factor constraint for clearing passengers at turn-back stations on short-turn routings;

$$L_z \leq L' \tag{9}$$

Since the train needs to clear out the passengers before turning back, the load factor of the turn-back train should not be too large.

6. Restriction on the number of routings.

$$\sum_{i=1}^{Q-1} \sum_{j=i+1}^{Q} X_{i,j} \leq m \tag{10}$$

Considering that the excessive number of routings will make the travel of passengers and the organization of trains difficult, the maximum number of routings is set to *m*, and *m* is between 1 and 5.

## 4. An Alternative-Routing-Set Based Algorithm for Solving Train Routing Scheme

First, the routing scheme is determined by the spatial imbalance coefficient. Second, all possible routings are generated according to whether the station has the line conditions for turning back, and on this basis, the alternative routings are obtained according to the principle that the short-turn routing should cover the section with a large section passenger flow volume. Next, calculate the passenger flow distribution matching of each routing and different routing combinations in time period *t*, and then get the routing scheme with the smallest deviation between transportation capacity and volume. Note that, to simplify the expression, $q_{e,\max}$ represents the maximum section passenger flow volume of the cross-section *e* in time period *t* in the algorithm. The specific calculation steps are as follows:

Step 1: If $\mu_e^s < 1.5$, adopt the scheme of full-length routing, and turn to Step 6; otherwise, adopt the scheme of full-length and short-turn routing, and turn to Step 2.

Step 2: According to whether the station has the line conditions for turning back, obtain the set of all possible routings by enumeration. Considering that the short-turn routing should cover the section with a large section passenger flow volume, get the alternative routing set. As shown in Figure 4, there are 4 turn-back stations, and the set of all possible routings is: $\{S_{1,2}, S_{1,3}, S_{1,4}, S_{2,3}, S_{2,4}, S_{3,4}\}$. However, $q_{2,\max}$ or $q_{5,\max}$ with a large section passenger flow volume is not covered by routing $S_{2,3}$, so the set of alternative routings is: $\{S_{1,2}, S_{1,3}, S_{1,4}, S_{2,4}, S_{3,4}\}$.

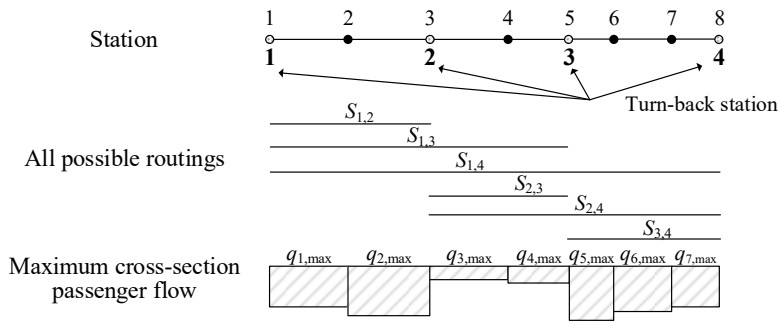

**Figure 4.** Schematic diagram of alternative routings.

Step 3: Calculate the passenger flow distribution matching of each routing in the set of alternative routings in time period *t*. The formula is as follows:

$$k_{i,j} = \begin{cases} -\dfrac{q_{i,j,\max}}{q'_{i,j,\max}}, \dfrac{q_{i,j,\max}}{q'_{i,j,\max}} < 1.5 \\ \dfrac{q_{i,j,\max}}{q'_{i,j,\max}}, \text{ otherwise} \end{cases} \tag{11}$$

$$\begin{cases} q_{i,j,\max} = \max(q_{e,\max}), \ e \in S_{i,j} \\ q'_{i,j,\max} = \max\left(q'_{e',\max}\right), \ e' \in S'_{i,j} \end{cases} \tag{12}$$

$$S'_{i,j} = \begin{cases} S_{j,j+1}, i = 1, j < Q \\ \{S_{i-1,i}, S_{j,j+1}\}, i > 1, j < Q \\ S_{i-1,i}, i > 1, j = Q \\ 0, i = 1, j = Q \end{cases} \tag{13}$$

where $k_{i,j}$ is the passenger flow distribution matching coefficient of routing $S_{i,j}$, $S'_{i,j}$ is the adjacent short-turn routing other than $S_{i,j}$, $q_{i,j,\max}$ is the maximum section passenger flow volume of routing $S_{i,j}$, $q'_{i,j,\max}$ is the maximum section passenger flow volume of routing $S'_{i,j}$, and $q'_{e',\max}$ is the maximum section passenger flow volume of the cross-section $e'$. Here, let the passenger flow distribution matching coefficient $k_{1,Q}$ of the full-length routing be 1.

Step 4: Calculate the passenger flow distribution matching under different routing combinations in the set of alternative routings in time period $t$. The formula of the evaluation parameters is as follows:

$$R = \sum_{i=1}^{Q-1} \sum_{j=i+1}^{Q} X_{i,j} \cdot k_{i,j} \tag{14}$$

where $R$ is the evaluation parameter for the matching of passenger flow distribution in time period $t$, and the larger $R$ is, the higher the matching degree between the transportation capacity and volume.

Step 5: Under the constraints of Equations (8)–(10), the CPLEX software is called to solve Equation (15), and the optimal train operation routing scheme with the best evaluation parameter $R$ is obtained. The formula is as follows:

$$\begin{aligned} &\min R \\ &\text{s.t.} \quad X_{1,Q} = 1 \end{aligned} \tag{15}$$

where $X_{1,Q} = 1$ means that the full-length routing is adopted.

Step 6: The passenger flow data is counted according to the obtained routing scheme, and on this basis, the number of trains running on each routing is calculated under constraints (4) to (7).

## 5. Case Study

Taking the actual data of the north section of Beijing Subway Line 8 in the morning peak hour as an example, the effectiveness and feasibility of the optimization method of the urban rail train operation scheme based on the control of the target load factor proposed in this paper are verified.

Line 8 (North Section) starts from Zhuxinzhuang Station in the north and ends at Zhongguo Meishuguan Station in the south. There are 19 stations, including 6 turn-back stations. Huilongguan Dongdajie Station, Yongtaizhuang Station, and Senlin Gongyuan Nanmen Station have the line conditions for turning back. The minimum turn-back time of Zhuxinzhuang Station is 120 s, and that of Zhongguo Meishuguan is 150 s. The line conditions of line 8 (North Section) are shown in Figure 5:

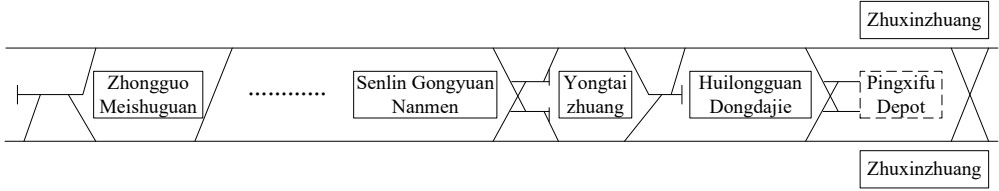

**Figure 5.** Schematic diagram of the line conditions of the north section of Line 8.

Based on the data of the time-sharing maximum section passenger flow volume on a certain day, the optimization of the train operation scheme was studied. The statistical results of the time-sharing maximum section passenger flow volume in the upward and downward directions are shown in Figure 6. Figure 6 shows the maximum section passenger flow data in the upward and downward directions collected every half hour. The abscissa represents different time periods. The statistical time ranges from 4:30 to 24:00. It

can be seen that the maximum section passenger flow volume for the whole day is in the period from 8:00 to 9:00 in the downward direction.

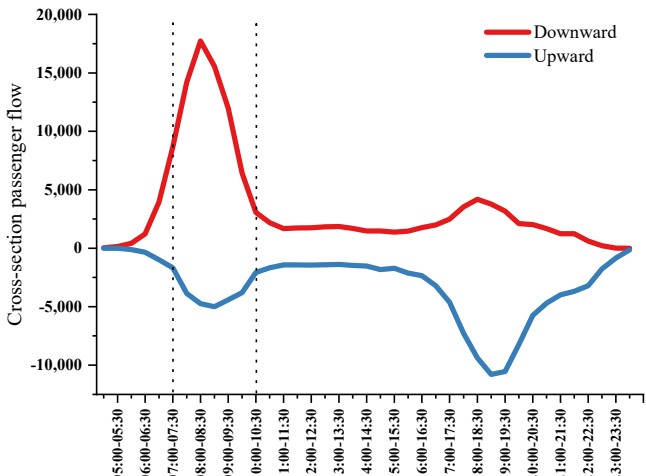

**Figure 6.** Time-sharing maximum section passenger flow volume in the upward and downward directions.

According to formula (1), the time imbalance coefficient of passenger flow is calculated (as shown in Figure 7). It can be seen from the figure that: (1) The time imbalance coefficients of the upward direction from 17:30 to 20:30 are all greater than 1.5. Therefore, it can be determined that the evening peak period in the upward direction is 17:30 to 20:30; (2) The time imbalance coefficients of the downward direction from 7:00 to 10:00 are all greater than 1.5. Therefore, it can be determined that the morning peak period in the downward direction is 7:00 to 10:00.

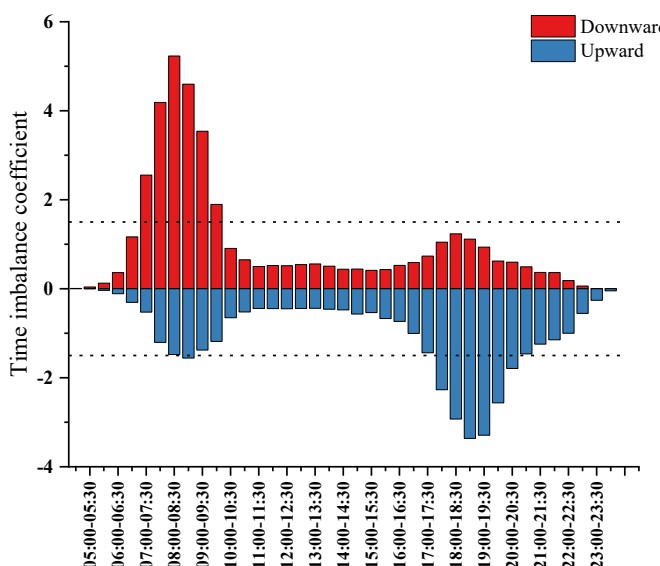

**Figure 7.** Time imbalance coefficient of passenger flow of the north section of Line 8.

The following conclusions can be drawn from the spatial imbalance analysis (as shown in Figure 8) during the morning peak period from 8:00 to 9:00 in the downward direction: (1) The passenger flow distribution of the section in the downward direction is imbalanced. The load factor of some sections during the peak hour is less than 30% (Zhuxinzhuang Station to Yuzhi Lu Station, and Gulou Dajie Station to Zhongguo Meishuguan Station), while that of some sections is greater than 90% (Lincuiqiao Station to Aolinpike Gongyuan Station);

(2) Generally speaking, the passenger flow of the section in the downward direction is relatively high, and there are 11 sections with a load factor exceeding 50% during the peak hour.

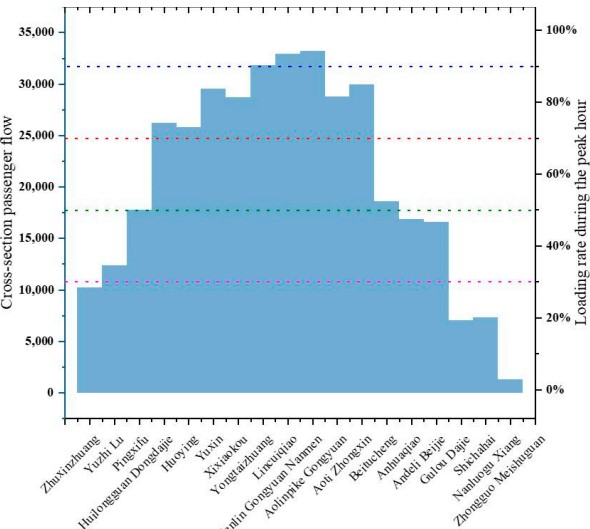

**Figure 8.** The largest section passenger flow volume in the morning peak.

According to Equation (2), the spatial imbalance coefficient of passenger flow in the downward direction during the morning peak period is calculated, as shown in Figure 9. Because the spatial imbalance coefficient from Yongtaizhuang station to Aolinpike Gongyuan Station is greater than 1.5, and that from Gulou Dajie station to Zhongguo Meishuguan Station is less than 0.5, the multi-routing scheme is adopted.

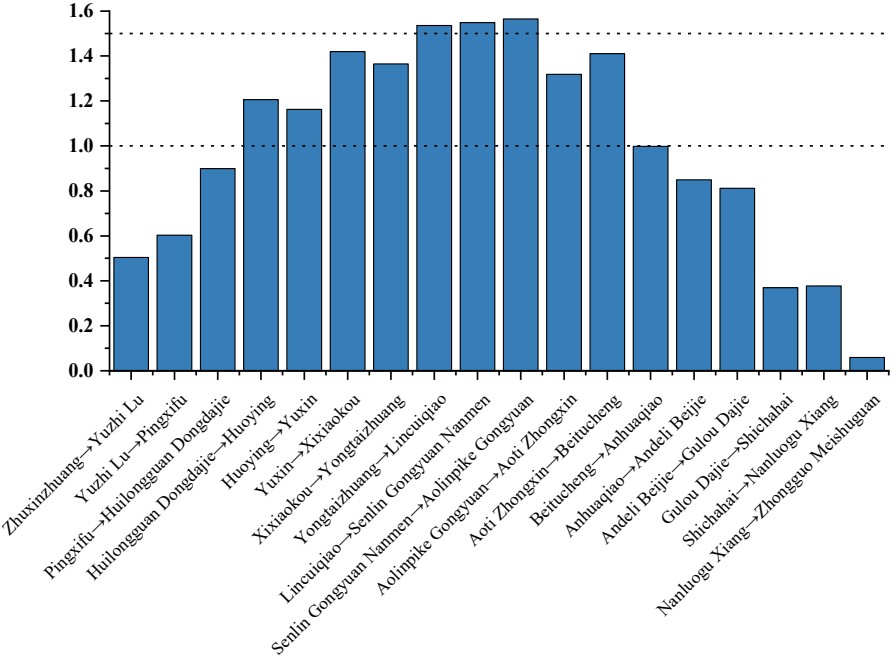

**Figure 9.** Spatial imbalance coefficient of downward passenger flow in the morning peak.

Based on the passenger flow data, the maximum cross-section load factor in the upward direction of Line 8 (North Section) is 27.6%. According to the operation experience, $L' = 50\%$, so the upward direction satisfies the requirement for the maximum load factor before turning back and clearing out passengers. Considering the turnaround conditions

of stations on the line, there are 10 routings in total, and the passenger flow distribution matching of each route during 8:00–9:00 is shown in Table 1. Here, the five stations with turn-back capability are numbered sequentially from 1. For example, $S_{1,2}$ means the routing from Zhuxinzhuang Station to Huilongguan Dongdajie Station. Combined with the maximum section passenger flow volume data, there are 9 alternative routings, and the set of alternative routing is $\{S_{1,3}, S_{1,4}, S_{1,5}, S_{2,3}, S_{2,4}, S_{2,5}, S_{3,4}, S_{3,5}, S_{4,5}\}$.

**Table 1.** Matching coefficient of passenger flow distribution of each route.

| Route Name | Matching Coefficient | Route Name | Matching Coefficient |
|---|---|---|---|
| $S_{1,2}$ | −0.60 | $S_{2,4}$ | −0.99 |
| $S_{1,3}$ | −0.90 | $S_{2,5}$ | 1.87 |
| $S_{1,4}$ | −0.99 | $S_{3,4}$ | −0.99 |
| $S_{1,5}$ | 1 | $S_{3,5}$ | −1.12 |
| $S_{2,3}$ | −0.90 | $S_{4,5}$ | −1.01 |

The evaluation parameter *R* of the passenger flow distribution matching of different combination schemes is shown in Figure 10. Combined with the constraints, the routings $S_{1,5}$ and $S_{2,5}$ (with an *R* value of 2.87) are selected, that is, the full-length routing from Zhuxinzhuang Station to Zhongguo Meishuguan Station, and the short-turn routing from Huilongguan Dongdajie Station to Zhongguo Meishuguan Station.

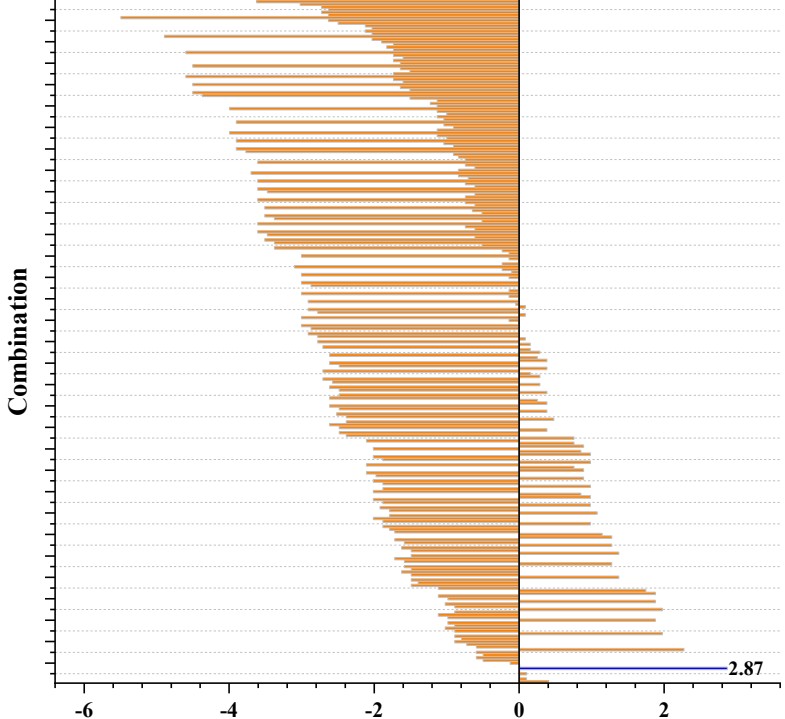

**Figure 10.** Evaluation parameters for matching of passenger flow distribution in different schemes.

The train capacity is 1468, with 80% as the control condition for the load factor of the train during peak hours. The section passenger flow volume in the upward direction is small, but to ensure the same service quality, the headway in the upward direction remains unchanged at 152 s. The optimization method of urban rail train operation scheme based on the control of the target train loading factor proposed in this paper was used to optimize the train operation scheme in the downward direction during the morning peak hours (7:00–9:00). According to the passenger flow data, the specific number of trains in the

non-collinear section and the collinear section under the condition of using routing $S_{1,5}$ and $S_{2,5}$ are:

$$17765 \div (1468 * 0.8) \approx 1535249 \div (1468 * 0.8) \approx 30$$

Therefore, the number of trains on routing $S_{1,5}$ is $k_1$, $k_1 = 15$, and the number of trains on routing $S_{2,5}$ is $k_2$, $k_2 = 30 - k_1 = 15$. For Line 8 (North Section), $K = 50$. Therefore, $k_1 + k_2 < K$, which satisfies the constraint of the total number of trains in use. The proportion of the full-length and short-turn routings is about 1:1. Line 8 (north section) has a minimum driving interval of 120 s for both the power supply system capability and the signal system capability. The relevant parameters before and after the optimization of the train operation scheme are shown in Table 2.

**Table 2.** Comparison table of parameters before and after optimization of train operation scheme (downward direction).

| Parameters | Original Scheme | Optimized Scheme |
|---|---|---|
| Minimum headway (s) | 152 | 120 |
| Maximum load factor for the collinear section of the full-length routing and short-turn routing | 101% | 80% |
| Maximum load factor for the non-collinear section of the full-length routing and short-turn routing | 51% | 80% |
| Proportion of the full-length and short-turn routings | — | 1:1 |

For the downward direction during the morning peak hour: (1) The minimum headway is shortened from 152 s to 120 s, the train capacity is increased by 21.1%, and the average waiting time of passengers is shortened by 16 s; (2) The proportion of the full-length and short-turn routings is 1:1; (3) The maximum load factor for the collinear section of the full-length routing and short-turn routing is reduced from 101% to 80%, which is a reduction of 21%; and (4) The maximum load factor for the non-collinear sections of the full-length routing and short-turn routing meets the specified load factor requirement of 80%, and the matching effect of capacity and volume is better. Therefore, under the constraints of the target load factor, the optimized cross-section's load factor in the downward direction is more balanced.

## 6. Conclusions

In this paper, under the constraint of line capacity, to achieve the goal of ensuring safe and orderly travel of passengers in the case of the ever-increasing passenger flow due to work resumption, an optimization model of the train operation scheme is proposed on the premise of load factor control. The passenger flow data of Line 8 (North Section) during the morning peak hours was used as an example for the analysis. By using the full-length and short-turn routings, the downward direction in the morning peak hours met the control requirement of the target load factor of 80%. The maximum load factor in the collinear section was reduced by 21%, and the capacity matching effect of the non-collinear section was improved because the maximum load factor reached the set load factor of 80%. The results show that the optimization method proposed in this paper is effective. Under the requirement of load factor control, the load factor of each section is more balanced after optimization, and the goal of optimal coupling of passenger flow and train flow was achieved.

In this paper, the express-local mode is not taken into account when optimizing the urban rail train operation scheme, which needs further study.

**Author Contributions:** Conceptualization, F.D., H.Y. and Y.W.; Data curation, H.Z. and Y.N.; Formal analysis, F.D. and H.Z.; Funding acquisition, F.D. and Y.N.; Investigation, F.D. and H.Y.; Methodology, F.D., H.Z. and H.Y.; Project administration, Y.N.; Resources, F.D. and Y.W.; Software, H.Z. and Y.N.; Supervision, Y.W.; Validation, F.D. and H.Z.; Visualization, F.D. and H.Z.; Writing—original draft, F.D. and H.Z.; Writing—review & editing, F.D. and H.Z. All authors have read and agreed to the published version of the manuscript.

**Funding:** This research was funded by Beijing Nova Program grant number Z211100002121098; Beijing Subway Innovation Research Project grant number 2022042103; National Key Technology Research, Development Program of China grant number 2020YFB160070X and the National Natural Science Foundation of China grant number 72001023.

**Informed Consent Statement:** Not applicable.

**Acknowledgments:** This work has been sponsored by Beijing Nova Program (Z211100002121098); Beijing Subway Innovation Research Project (2022042103); National Key Technology Research, Development Program of China (2020YFB160070X) and the National Natural Science Foundation of China (72001023).

**Conflicts of Interest:** The authors declare no conflict of interest.

## Nomenclature

| | |
|---|---|
| $C$ | train capacity [person] |
| $L_t$ | predetermined load factor in time period $t$ [%] |
| $N_t$ | number of trains in time period $t$ |
| $h_t^s$ | headway in time period $t$ in direction $s$ [second] |
| $h_{\max}$, $h_{\min}$ | maximum headway and minimum headway [second] |
| $T$ | statistical period |
| $h'_{\min}$ | minimum headway that the power supply system can support [second] |
| $h''_{\min}$ | minimum headway that the signal system can support [second] |
| $L_{\max}$ | target value of the load factor [%] |
| $k_r$ | number of train pairs for routing $r$ |
| $K$ | total number of trains |
| $S_{\text{start}}$, $S_{\text{end}}$ | starting and ending locations of the short-turn routing |
| $N$ | total number of stations on the line |
| $L_z$ | load factor of the turn-back train [%] |
| $L'$ | maximum allowable load factor before turning back and clearing out passengers [%] |
| $Q$ | number of stations with line conditions for turn-back operation |
| $S_{i,j}$ | routing from turn-back station $i$ to turn-back station $j$ |
| $q_{t,\max}^s$ | maximum section passenger flow volume in time period $t$ in direction $s$ [person] |
| $q_{e,\max}^s$ | maximum section passenger flow volume of cross-section $e$ in direction $s$ [person] |
| $E$ | number of sections |
| $X_{i,j}$ | 0–1 variable representing the routing selection. If $S_{i,j}$ is used, $X_{i,j} = 1$, otherwise $X_{i,j} = 0$. |

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
