# Peer review of "An Optimization Method of Urban Rail Train Operation Scheme Based on the Control of Load Factor"

_sustainability, doi:10.3390/su142114235_

Round 1

Reviewer 1 Report

More references should be cited and a comprehensive literature review is required

The disadvantage of this work should be discussed and future works or directions should be presented;

I want to know what is the engieering applications or significance of this work. Would the proprosed model be used in operations or how to modify the model if it is used to control opreation.

Reviewer 2 Report

In the abstract, it is suggested to highlight the novelty of this work, as the optimisation of the train schedule has already been extensively studied. Which problem is targeted by this paper?

In the first paragraph of the introduction, it is recommended to give each object a reference example for the authors to refer to. For instance, ‘… under the constraint of line signal system capacity [1], power supply system capacity [2], turn-back capacity [3] and so on…’

[1] Goverde, Rob MP, Francesco Corman, and Andrea D’Ariano. "Railway line capacity consumption of different railway signalling systems under scheduled and disturbed conditions." Journal of rail transport planning & management 3.3 (2013): 78-94.

[2] Y. Song, Z. Liu, A. Ronnquist, P. Navik, Z. Liu, Contact wire irregularity stochastics and effect on high-speed railway pantograph-catenary interactions, IEEE Trans. Instrum. Meas. 69 (2020) 8196–8206.

[3] Shi, Haiou, et al. "Study on the Optimal Safety Distance for Turn-back Line of Regional Railway." IOP Conference Series: Materials Science and Engineering. Vol. 688. No. 4. IOP Publishing, 2019.

In the second paragraph, the works in [1-5] should be further discussed, which yields the motivation for this work.

It is recommended to further explain the shortcoming in previous research. What kind of assumption and unspecific constraints limit the use of previous optimisation approach for scenarios with target train full loading rate during major emergencies? Please expand the description to clarify the scientific challenge targeted by this work.

The analysis of passenger flow relies on the acquisition of operational data. Please specify how this data is acquired.

A more concise title is recommended for Section 4.

Please specify what kind of optimisation approach is used here and what is the reason for the selection.

Figure 6 is almost unreadable. Please clarify the meaning of the x-axis.

The conclusions can be improved to highlight what kind of shortcoming in previous research is solved instead of just describing the work in this paper.

Round 2

Reviewer 2 Report

All my comments have been well addressed by the authors. I recommend the publication of this paper.

Author Response

We have polished the article according to the experts' suggestions. Thank you again for your valuable comments. 

Reviewer 3 Report

The paper has been significantly improved and some of my concerns have been addressed.

Can the authors provide some evidence about how close to optimal are the results?

I still don't think that the "capacity" of any component should be measured in time units. The term "capacity" should probably be replaced here with "turnback time" or a related term. 

"Cross section passenger flow" should probably be repaced with "passenger flow diagram".

From the responses to comments: "The “routing mode” here refers to the specific routing mode adopted by the train" seems a circular non-explanation. Also, in the transportation literature “mode” means a different form of transportation.

“Travel interval” should be replaced with headway.
